# Liquid-Metal-Based Magnetic Controllable Soft Microswitch with Rapid and Reliable Response for Intelligent Soft Systems

**DOI:** 10.3390/mi13122255

**Published:** 2022-12-18

**Authors:** Qin Jiang, Zhitong Hu, Yaping Xie, Kefan Wu, Shuo Zhang, Zhigang Wu

**Affiliations:** State Key Laboratory of Digital Manufacturing Equipment and Technology, School of Mechanical Science and Engineering, Huazhong University of Science and Technology, Wuhan 430074, China

**Keywords:** liquid metal, microswitch, magnetic controlling, soft robots, cohesion force

## Abstract

When combined with diverse sensors, soft robots significantly improve their functionalities and intelligence levels. However, most of the existing soft sensors require complex signal analysis devices or algorithms, which severely increase the complexity of soft robot systems. Here, based on the unique fluidic property of liquid metal, we propose a magnet-controllable soft microswitch that can be well-integrated into a soft robot system, e.g., a soft gripper to help it facilely detect and precisely grab objects. The microswitch consists of a flexible soft beam electrode and a fixed electrode, forming a soft microsystem. By tuning the cohesion force of the liquid metal between the electrodes, the microswitch can convert its states between an individual and a self-locking state. The microswitch can achieve a reasonable rapid response (~12 ms) and high switching frequency (~95 Hz). Furthermore, soft microswitches can be customized into logic units and also coupled to control a digital tube showing various numbers. Our work provides a new simple soft sensor unit that may enhance the intelligence of soft systems.

## 1. Introduction

Soft robots have been greatly developed over the past decades due to their unique compliance, natural deformability, and safe interactivity. They can be employed in a wide range of unconstructive/unpredictable scenarios [1], e.g., the grabbing of tender objects [2,3,4,5], invasive surgery [6,7], and a human–machine interface [8,9]. To adapt to diverse and ever-increasingly complex scenarios, more and more soft sensors are being integrated into soft robots to improve their sensibility and intelligence [10], such as tactile sensors [11,12], optical sensors [9,13], and magnetic sensors [14]. However, most soft sensors output continuous analog signals, including voltage, resistance, capacity, or inductance. Usually, these soft sensors require sophisticated algorithms, expensive signal detective devices, and cumbersome data pre-processing, significantly increasing the complexity of the soft robot system. Meanwhile, due to their simple structures and stable response, mechanical contact switches are widely used in traditional rigid robot systems. These switches output digital signals with a high/low level, greatly simplifying the signal processing. Hence, developing such soft switch may provide a new way for the intelligence development of soft robots.

Owing to its high conductivity, high fluidity, and ultra-stretchability [15], liquid metal (LM, e.g., a gallium-based alloy) is widely utilized in the fabrication of diverse soft sensors, for example, strain sensors [16,17], haptic sensors [18,19], magnetic sensors [20], and pressure sensors [21,22]. While most of these soft sensors utilize LM as flexible conductive wires or particles, there has been relatively little exploration into its unique feature, the fluidic digital connectivity of the LM. Although several examples of LM-based electrical switches have been demonstrated, these switches need specific working environments (for example, NaOH [23,24,25] or HCL [26] solution immersion), severely restricting their application scenarios.

Here, we present an LM-based magnetic controllable microswitch (MCSM), which can be well-integrated into a soft system; a gripper is used to help it precisely detect and fit various grab objects of different shapes and grab them automatically without external triggering or disturbing the central control/data processing system (Figure 1). When the soft gripper contacts a target object, the soft beam contactor connects with the fixed contactor under magnetic attraction. In that case, the MCSM transfers a digital contact signal to the microcontroller, controlling the soft gripper to inflate and grab objects. Using such a simple feedback strategy, the soft gripper possesses a self-sensing capability and can replace/simplify the traditional complex external detection system and data processing unit.

## 2. Materials and Methods

### 2.1. Working Principle

As a core component of MCSMs, the soft beam can be either in connection or non-connection state under the magnetic field control. Thus, in order to achieve stable control of MCSMs, it is vital to investigate appropriate design parameters and working principles.

As shown in Figure 2a, a soft beam with an LM electrode is fabricated by a UV laser to remove the superfluous LM film. Owing to the high machining precision of the laser beam, the untreated soft electrode shows a smooth surface. Being attracted by an under-placed magnetic field, the soft beam’s bending angle (*θ*) is associated with the beam thickness (Figure 2b). As the thickness increases from 250 μm to 450 μm, the bending angle decreases from 74.9° to 51.5°, whereas the bending stiffness of the soft beam increases from 0.98 μN∙m to 5.73 μN∙m. The simulation result indicates that the soft beam’s bending status can be tuned by an applied magnetic field, which is generated by an electromagnet bellowing (Figure 2c).

When the soft beam electrode bends to contact with the fixed electrode, the LM between these two electrodes facile connects and integrates under the action of its surface tension and metallic bonding [27]. In the meantime, the solid oxide skin of the LM instantaneously generates and partially enhances the bridge strength along the increment of the new outer surface of the liquid bridge. Therefore, the cohesion force, $F_{coh}$, of the LM exists between these two electrodes, impeding them from separating. We investigate the maximum separation force, $F_{sep}$, of two electrodes on different spray times of LM (Figure 2d and Figure A1). As the spray time of LM increases, the maximum separation force, $F_{sep}$, gradually increases and stabilizes after a spray time of 1.25 s. Moreover, the thickness of LM film has the same trend with increased spray times (Figure A1). Therefore, the separation force, $F_{sep}$, is positively correlated with the LM’s film thickness. We record the generation and breakdown of the liquid bridge of LM during the slow separation of two electrodes (Figure 2e and Appendix A).

By combining different LM spray times and soft beam thicknesses, we can configure two types of MCSMs: (I) non-self-locking type, where the $F_{re}$ is higher than the $F_{coh}$, and, thus, the soft beam can spring back; (II) self-locking type, where the $F_{re}$ is less than the $F_{coh}$, and, thus, the soft beam is stuck on the fixed electrode (Figure 2f). Combined with the unique rheological behaviors of LM, we develop a simple design paradigm to achieve state conversion via MCSMs.

### 2.2. Fabrication Process of the MCSMs

#### 2.2.1. Materials Preparation

Two types of soft substrates are used in the MCSM, carbon-doped PDMS (cPDMS) and magnetic PDMS (mPDMS), separately. The cPDMS was prepared by mixing native PDMS (Sylgard 184, Dow Corning Corporation, Midland, MI, USA) and carbon black (XC72R, CABOT, Alibaba, Hangzhou, China) in a ratio of 10:1:0.2 (silicon base/curing agent/carbon black). The mPDMS was prepared by mixing PDMS and NdFeB particles (MQFB-B-20076-089, Magnequench, Indianapolis, IN, USA) in a ratio of 20:1:20 (silicon base/curing agent/NdFeB particles). Both mixing procedures should be well-mixed (at least 15 min) for uniform particle distribution in PDMS. After bubbles’ removal, the mixture was filmed on an aluminum plate using a thin-film applicator to control the film’s thickness. mPDMS substrates were scraped for different thicknesses (250, 350, and 450 μm), and cPDMS substrates were scraped for 500 μm. Then, these substrates were cured at 90 °C for 45 min in an oven (UF 55 plus, Memmert, Schwabach, Germany). The mPDMS substrates were magnetized under a ~2.4 T pulsed magnetic field device (MAG-3000, CH-Magnetoelectricity Technology Co., Ltd., Hangzhou, China).

#### 2.2.2. Fabrication Process of the MCSM

The fabrication can be summarized into four parts, as shown in Figure 3.

Spraying of the LM. The LM (Galinstan, 68.5% Ga, 21.5% In, 10% Sn; Geratherm Medical AG, Geschwenda, Germany) film was atomized sprayed on the surface of mPDMS and cPDMS substrates, using a spray gun (Meec tools; Jula, Sweden). The spray time (from 0.3 to 2 s) and air pressure (0.35 Mpa) can be regulated by a pressure regulator (ML-5000XII; Musashi Engineering, Tokyo, Japan). The distance between the nozzle and substrate is about 5 mm.Laser treating. A UV laser marker (HGL-LSU3/5EI, Huagong Laser, Wuhan, China, wavelength: 355 nm, pulse repetition frequency: 50 kHz) was utilized to treat three different layers. Each layer was treated in two steps. The first step aimed to remove the superfluous LM film and retain the LM patterns. Then, the second ablation step with higher energy aimed to cut substrates, forming a beam and hollow. For the first step, a laser with a scanning speed of 400 mm s^−1^, an adjacent distance of 0.05 mm, and a pulse width of 0.1 μs was used for the LM removal. For the second step, a laser with a moving speed of 40 mm s^−1^ and a pulse width of 0.2 μs was used for substrate cutting.Assembling. Three soft layers were aligned layer by layer and bounded by the uncured PDMS. Then, the MCSM was placed in the oven at 90 °C for 25 min. to bound well.Working. A micro-switch was placed on the electromagnet. Under the attraction of the magnetic field, the MCSM is turned on.

#### 2.2.3. Methods of Verification and Demonstrations

*Characterizations:* The surface of the soft beam was captured by an ultra-depth 3D microscope (DSX 510, Olympus, Pennsylvania, PA, USA). The surface morphologies of the beam surface were characterized by an SEM (FSEM, GeminiSEM300, Carl Zeiss, Wetzlar, Germany). The capillary bridge (Figure 2e) was recorded by a drop-shape analysis apparatus (DSA25, KRUSS, Humburg, Germany). The bending stiffness of the soft beam was calculated by $Et^{3}/12\left(1-v^{2}\right)$, where *E* is Young’s modulus, *t* is the thickness, and *v* is the Poisson ratio of the mPDMS. *E* is 0.57 MPa (obtained by uniaxial tensile tests), and *v* is set as 0.49.

*Measurement of the separation force of LM electrodes:* The details of the separation force measurement can be seen in Figure A2.

*Numerical simulation*: The simulation was conducted with COMSOL 5.6a. (COMSOL Inc., Stockholm, Sweden) The Yeoh hyperelastic model was selected to simulate the incompressible silicone rubber beam (PDMS).

*Demonstrations*: An electromagnet (YHN-P08, 8 mm diameter, Alibaba) was placed below the MCSM and separated by a glass slide (1 mm thickness). A magnetometer (TD8650, TUNKIA, Shenzhen, China) was used to measure the soft beam’s magnetic induction intensity at ~40 mT. The soft gripper was molded from two kinds of silicone: a stretchable layer and a limiting layer, which were from Ecoflex 00-30 and Ecoflex 00-50 (Smooth-On, Pennsylvania, PA, USA), respectively. Then, the obtained two layers were bonded by a silicon rubber adhesive (Sil-poxy; Smooth-On, Smooth-On, Pennsylvania, PA, USA). The magnetic rubber sheet (10 mm diameter and 1 mm thickness) was fabricated by mPDMS.

## 3. Results

### 3.1. Electrical Characterization

To characterize the electrical properties of the MCSM, we conducted a series of experiments, as shown in Figure 4. We used an electromagnet to actuate the MCSM and record the variation of the voltage on the electromagnet (U_1_) and the voltage on the constant resistance (U_2_) over time, as shown in Figure 4a. After U_1_ rises to ~7 V, the electromagnet can generate enough magnetic force to turn the MCSM ON. Then, the circuit is connected, and U_2_ rises to ~1.5 V. As shown in the zoomed-in view, the delay time between U_1_ and U_2_ is about 12 ms, indicating that the MCSM has a rapid response.

Next, the switching frequency of the MCSM was investigated, as shown in Figure 4b. At first, the MCSM can keep pace with the increasing magnetic frequency until 95 Hz. When the magnetic frequency reaches 100 Hz, the MCSM cannot work well and shows a “step-out” effect. Under this highly magnetic frequency, the soft beam vibrates with too weak an amplitude to connect with the fixed electrode. Therefore, the MCSM can reach a high switching frequency (~95 Hz), accommodating most low-speed circuit-controlling scenarios.

We also investigated the repeat performance of the MCSM using a mechanical cycling test (Figure 4c and Figure A3). The MCSM is connected in series with a constant resistor (150 Ω) that is much higher than its own resistance. During the 1500 cycles of ON/OFF converting by the MCSM, the whole resistance of the circuit remains stable, Figure 4c. With the oxidation of LM gradually increasing, the resistance of the MCSM rises slightly, from 1.243 to 1.277 Ω. Owing to the excellent connectivity and fluidity of the LM, the electrodes of the MCSM can stand long times of connection and separation.

As a core controlling component in circuits, the maximum loading current of the MCSM is essential. Figure 4d shows the working state of the MCSM with various loading currents and voltages. As the loaded current linearly rises from 0.28 to 2 A, the MCSM can keep the electrical connection. However, when the loaded current is higher than 2 A, the LM electrode of the soft beam generates a break zone, resulting in the MCSM being disabled (Figure A4). The main reason for this phenomenon is the electromigration of the LM film under this high current [28,29,30,31]. The result shows that the MCSM can stand a high loaded current (~2 A), which indicates its potential in most scenarios of signal delivery.

### 3.2. Demonstrations

Several demonstrations of MCSMs for soft electronics and soft robots are shown in Figure 5. In Figure 5a, we demonstrate three basic logic units (AND, OR, and NOT gate) by combining MCSMs with electromagnets. Unlike conventional logic circuits that use diodes or transistors for signal transferring or blocking, we used interactions between the magnetic field and the soft beam to achieve the same effect. We define that V_m represents the input state of the controlling electromagnet: the Symbols (1) and (−1) of V_m both present that the electromagnet is powered on, yet the directions of its loaded voltage and generation magnetic fields are opposite. When V_m is set to Symbol (1), the magnetic field will attract the soft beam to make the MCSM connection. When V_m is set to Symbol (−1), the magnetic field will generate a repulsive force to turn off the MCSM. Symbol (0) of V_m indicates that the electromagnet is powered off, so the magnetic field disappears. Moreover, V_out represents the output state of the gate: Symbol (1) represents that the gate outputs a high-level signal; Symbol (0) represents that the gate outputs a low-level signal.

The AND gate is designed by two non-self-locking MCSMs in a serial connection. Only when double electromagnets input Symbol (1) will the AND gate output Symbol (1). The OR gate is assembled by two non-self-locking MCSMs in parallel connection. If there is more than one input present for Symbol (1), the output will be Symbol (1). Particularly, the NOT gate is designed by a self-locking MCSM to close normally. Initially, the input of V_m is set to Symbol (1) to close the MCSM, so the gate outputs Symbol (1). Then, even if V_m is set to Symbol (0), the MCSM stays closed, while the NOT gate continues to output Symbol (1). However, when the input of V_m converts to Symbol (−1), the MCSM turns OFF via the reverse magnetic field, and the NOT gate outputs Symbol (0). This demonstration shows the application potential of the MCSM in the logical control of and computation for soft robots.

We also utilize seven MCSMs to control a digital tube for showing various numbers, as shown in Figure 5b. In the circuit-connection diagram, each MCSM controls a pin of the digital tube (Figure A5). Therefore, by turning different MCSMs ON/OFF, the diagram shows the different diodes of the digital tube light and various numbers. This demonstration shows the potential of the MCSM to achieve large-scale circuit control in future soft robots.

Furthermore, the fully soft MCSM can be well-integrated into a pneumatic-actuated soft gripper, as shown in Figure 5c (the setup details are shown in Figure A6). As shown in the schematic, there is a permanent magnetic rubber sheet at the top of the MCSM. In the initial state, the magnetic force of the rubber sheet is insufficient to attract the soft beam, owing to the gap between them. After a finger presses the rubber sheet to deform and close the MCSM, the soft beam electrode is attracted under the magnetic field and connects with the fixed electrode. Then, the microcontroller receives a high-level signal and controls the air pump to inflate the soft gripper. Due to the simple form of the transmitted signal, complex signal-processing algorithms are not required for the microcontroller, which significantly improves the response speed of the soft gripper. With the help of the MSCM, the soft gripper can automatically and precisely detect and fit various grabbed objects with different shapes, as shown in Figure 5d. The MCSM provides a highly efficient and convenient self-sensing paradigm to enhance the intelligence of soft robots.

## 4. Conclusions

In this study, by tuning the separation force between electrodes and combining it with the unique rheological behaviors of LM, we present an MCSM that can be well-integrated into soft systems to level up their intelligence. The MCSM offers several favorable characteristics: fast response (~12 ms), high switching frequency (~95 Hz), reliable durability (over 1500 cycling times), and high loading current (~2 A). Several demonstrations have been presented to explore their potential. Configured by different types of MCSMs, several basic logic units (AND, OR, and NOT gates) are presented to show MCSMs’ potential for logic and further computation. Simultaneously, seven MCSMs can combine and control a digital tube to show different numbers. Finally, integrating the MCSM allows for a soft gripper to facilely detect and precisely grab objects, using a stable and convenient feedback-control strategy. Our approach offers a new solution for developing a soft, simple, and stable sensor unit, which can potentially promote the development of soft intelligent systems.

## Figures and Tables

**Figure 1 micromachines-13-02255-f001:**
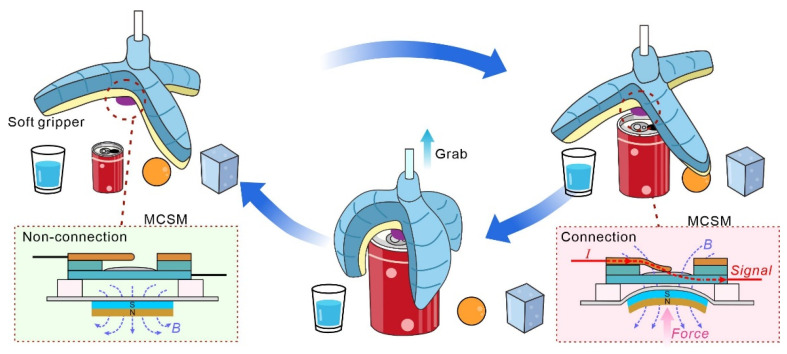
Schematic of a soft gripper integrated with an MCSM to self-sensing detect and grasp objects.

**Figure 2 micromachines-13-02255-f002:**
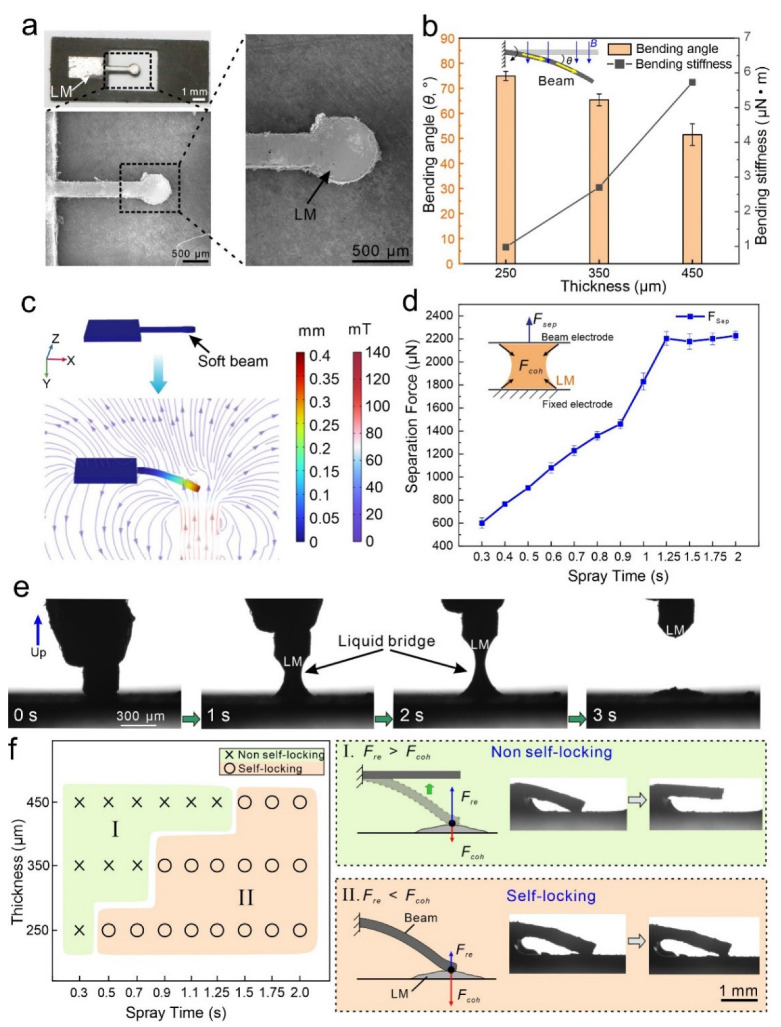
Working principle. (**a**) SEM photos of the beam electrode. (**b**) Bending angle and bending stiffness of the soft beam with different thicknesses. (**c**) Numerical simulation of the soft beam bending status under a magnetic field. (**d**) Maximum separation force, $F_{sep}$, of electrodes with different spray times. (**e**) Serial photos of the liquid bridge during the slow separation of electrodes. Separation speed = 1.5 mm/s. (**f**) Two states of the MCSM produced by different spray times and thicknesses of the soft beam, where $F_{re}$ is the resilience force of the soft beam, and $F_{coh}$ is the cohesion force between two electrodes.

**Figure 3 micromachines-13-02255-f003:**
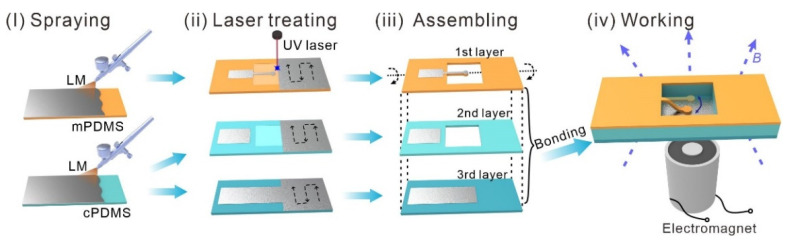
Fabrication process of the MCSM.

**Figure 4 micromachines-13-02255-f004:**
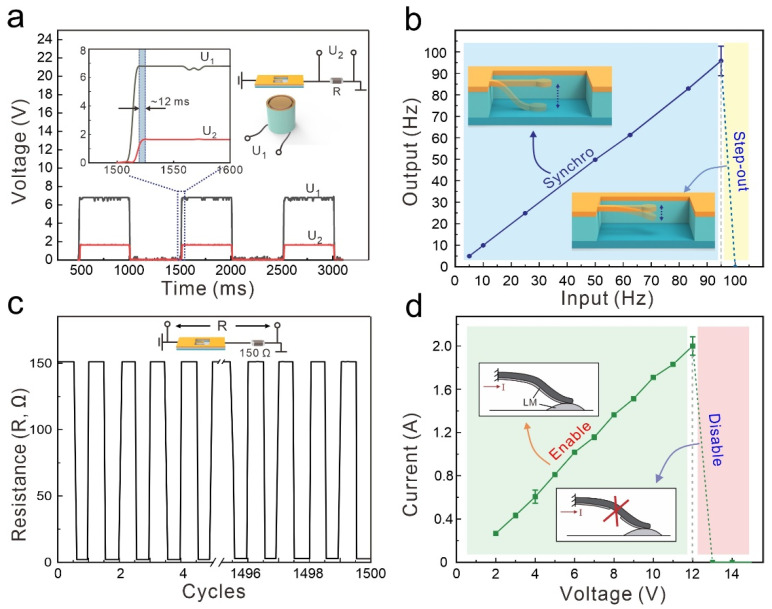
Electrical characteristics of MCSM. (**a**) The voltage on the electromagnet (U_1_) and the constant resistance (U_2_) at different times. (**b**) Connection frequency of a soft microswitch under different actuation frequencies. (**c**) Measured resistance (R) of the circuit during 1500 times cycling. (**d**) Loaded current of the microswitch under different voltages.

**Figure 5 micromachines-13-02255-f005:**
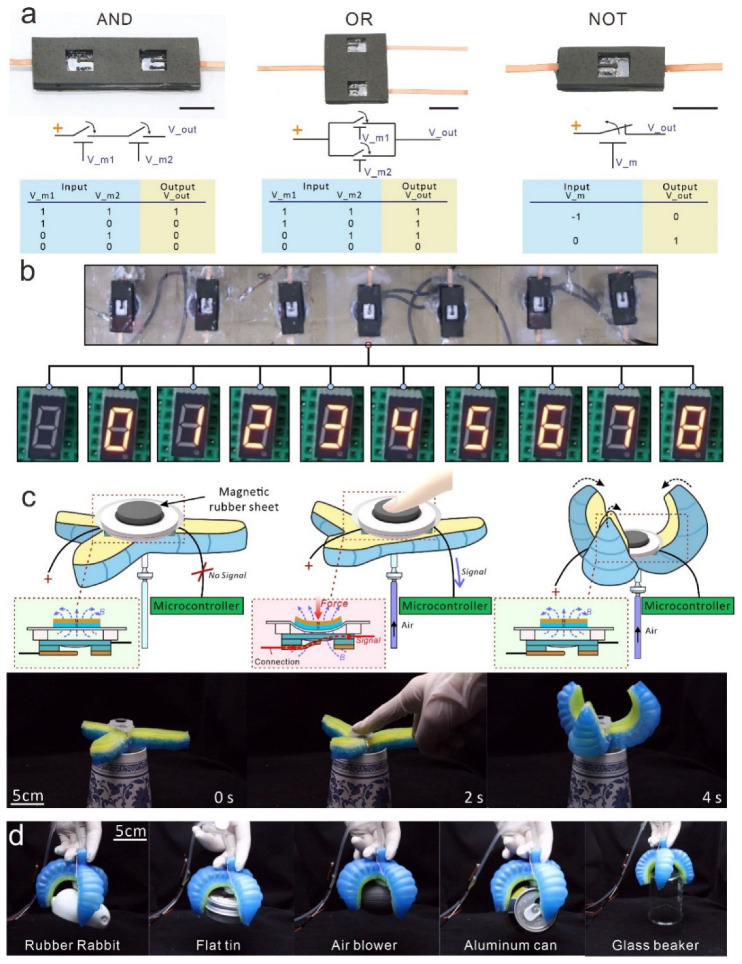
Demonstrations. (**a**) Logical units and truth tables with various MCSMs’ and electromagnets’ configurations. Scale bars = 5 mm. (**b**) A digital tube controlled by seven MCSMs shows various numbers. (**c**) Serial schematic and photos of a soft gripper contraction after a finger touches the MCSM. (**d**) Serial snapshots of the soft gripper integrated with an MCSM detecting and grabbing diverse objects.

## Data Availability

The datasets generated during and/or analyzed during the current study are available from the corresponding author on reasonable request.

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
