# Peer review of "Liquid-Metal-Based Magnetic Controllable Soft Microswitch with Rapid and Reliable Response for Intelligent Soft Systems"

_micromachines, 2022, doi:10.3390/mi13122255_

Round 1
Reviewer 1 Report
This paper presents a study on creating a mechanical switch that is based on a contact between a soft beam with liquid metal. The design and manufacturing method are presented. Validation experiments and characterization results on the mechanical and electrical properties ar represented. Experimental videos showing the liquid metal motion contact mechanism and experimental demonstrations are provided.
#Major comments#
The paper represents the idea in the introduction well enough. But the experimental descriptions in the material and methods, and also the results section, are difficult to understand. There is much information tangled each other. The readers lose themselves because there is no reasoning on why these analyses are being conducted and what is being learned from those experiments.
1. Section 3.1 (Working Principle of the MCSM) section should go under Materials and Methods because the working principle should be the very first thing that has to be explained under the materials and methods section. It has to provide how this mechanism is designed, how it functions, and what kind of sense is integrated. Then, the numerical computations during the tests should be provided in the Results section along with Figure 2.
2. In line 157, the authors mention the response is 12 ms, so it is instant. The word instant is overclaiming here. It is not instant in the perspective of electrical circuits because electrical connections and analog signals respond in a much faster fashion. In such a mechanical system, this time is limited by the motion of the soft beam motion. Since it is a mechanical system, this takes 12 ms, and the authors can not call this an instant. The wording has to be softened to avoid overclaiming.
#Minor Comments#
1. The word 'electromagnetic' is an adjective, not a noun. The authors repetitively use this word as a noun, which is a basic grammar mistake in this paper. For example, on page 3, line 94, and page 5, line 152.
2. In Line 153, the word electromagnetic should be 'voltage on the electromagnet' because there is no such thing as 'electromagnetic voltage'.
3. In line 153, again, the word 'resistance voltage' is very ambiguous. It can be replaced as 'Voltage on the substrate due to touch connection'.
4. In line 172, 'As the loaded current linear rises from ...' should be replaced by 'As the loaded current linearly rises from ...'
Reviewer 2 Report
This manuscrpt presents a magnetic control soft swith, which ultilizes the phase transition of Liquid-metal to Lock and release the switch contacts. The research is very interesting. Programmable flexible switches have broad application prospects in the field of flexible stretchable devices. However, the reviewers believe that there are still many areas for improvement before this article is accepted. Here are some revision suggestions:
Major issues:
1. The introduction part need to be reorganized. Readers can accurately understand the motivation of the article and the development of the field through the introduction. It is highly recommended to add some related researches, e.g.,
Magnetic control flexible switch:[1]A shape-deformable liquid-metal-filled magnetorheological plastomer sensor with a magnetic field ‘‘on-off’’ switch. iscience, 2022. [2] Magnetic Shape Memory Polymers with Integrated Multifunctional Shape Manipulation. Advanced Materials, 2018. (This ref. proposes a lockable soft switch)
Self-locking structure base on liquid-metal:[1] Liquid-Metal Magnetic Soft Robot With Reprogrammable Magnetization and Stiffness. IEEE RAL, 2022.(This ref.proposes a shape fixation method based on the phase transition of liquid metal)
2. For Fig.1, the reviewer suggested the preparation method as a separate drawing.
3. The structure of this manu should be reorganized. e.g., part 3.1 as the working principle should be placed before the preparation process (part 2). Part 3 should only discuss the experimental design and results.
4. As we all know, when gallium is exposed to air, it will be oxidized quickly to produce Ga2O3. This change will affect the conductivity of the liquid metal wire. Therefore, the electrical connection of the contact may fail. How the author will deal with this problem?
5. Fig.4 proposed some logic switch as application cases. However, in principle, we cannot build a NOT gate/OR gate circuit by only relying on mechanical switches. e.g., as shown in Fig. 4a, the non logic here is realized by switching action, rather than the level inversion logic in the real sense.
To realize NOT logic through analog circuit, at least two components with unidirectional conductivity (such as diode) should be required, and mechanical switch obviously does not have single conductivity. Therefore, the experimental description here should be treated with more caution.
Minor issues:
1. In fig.1b(1) This process should be "flame plating" or just "sparying".
2. In fig.1b(2) It is suggested use the description "Laser cutting ".
3. In fig.2a,Please add the scale to the picture.
4. In fig.4b, The "ON" and "OFF" states of the switch cannot be observed from the top view.
Reviewer 3 Report
The readability of the publication could be improved, the pictures could be assigned directly to the text. The reader must first search for the associated images and must jump to the next page to do so. Disturbs the reading flow.
Information on particle distribution in the PDMS is missing and thus also statements on reproduction.
The additional images should either be included in the actual publication. This way it appears as a loose supplement. If of importance then please include directly.
Line 175 refers to electromigration as the cause. Are there any images here that show this or is this a hypothesis?
Round 2
Reviewer 2 Report
The author revised the article according to the suggestions of the reviewers. It is recommended to accept this article.